# Possibility of Cell Block Specimens from Overnight-Stored Bile for Next-Generation Sequencing of Cholangiocarcinoma

**DOI:** 10.3390/cells13110925

**Published:** 2024-05-28

**Authors:** Mitsuru Okuno, Tomohiro Kanayama, Keisuke Iwata, Takuji Tanaka, Hiroyuki Tomita, Yuhei Iwasa, Yohei Shirakami, Naoki Watanabe, Tsuyoshi Mukai, Eiichi Tomita, Masahito Shimizu

**Affiliations:** 1Department of Gastroenterology, Gifu Municipal Hospital, Gifu 500-8323, Japan; keisukeiwata@nifty.com (K.I.); festinalenteyu@gmail.com (Y.I.); tsuyomukai@yahoo.co.jp (T.M.); etomita_jp@yahoo.co.jp (E.T.); 2Department of Gastroenterology, Matsunami General Hospital, Gifu 501-6062, Japan; 3Department of Tumor Pathology, Gifu University Graduate School of Medicine, Gifu 501-1193, Japan; kanayama.10mo16@gmail.com (T.K.); tomita.hiroyuki.y6@f.gifu-u.ac.jp (H.T.); 4Department of Diagnostic Pathology, Gifu Municipal Hospital, Gifu 500-8323, Japan; tmntt08@gmail.com (T.T.); naoki@watanabe.name (N.W.); 5Center for One Medicine Innovative Translational Research, Gifu University Institute for Advanced Study, Gifu 501-1193, Japan; 6First Department of Internal Medicine, Gifu University Hospital, Gifu 501-1112, Japan; shirakamiyy@yahoo.co.jp (Y.S.); shimim-gif@umin.ac.jp (M.S.); 7Department of Gastroenterological Endoscopy, Kanazawa Medical University, Ishikawa 920-0265, Japan

**Keywords:** bile cell block, surgically resected specimen, cholangiocarcinoma, next-generation sequencing, genetic alterations

## Abstract

The identification of anticancer therapies using next-generation sequencing (NGS) is necessary for the treatment of cholangiocarcinoma. NGS can be easily performed when cell blocks (CB) are obtained from bile stored overnight. We compared NGS results of paired CB and surgically resected specimens (SRS) from the same cholangiocarcinoma cases. Of the prospectively collected 64 bile CBs from 2018 to 2023, NGS was performed for three cases of cholangiocarcinoma that could be compared with the SRS results. The median numbers of DNA and RNA reads were 95,077,806 [CB] vs. 93,161,788 [SRS] and 22,101,328 [CB] vs. 24,806,180 [SRS], respectively. We evaluated 588 genes and found that almost all genetic alterations were attributed to single-nucleotide variants, insertions/deletions, and multi-nucleotide variants. The coverage rate of variants in SRS by those found in CB was 97.9–99.2%, and the coverage rate of SRS genes by CB genes was 99.6–99.7%. The NGS results of CB fully covered the variants and genetic alterations observed in paired SRS samples. As bile CB is easy to prepare in general hospitals, our results suggest the potential use of bile CB as a novel method for NGS-based evaluation of cholangiocarcinoma.

## 1. Introduction

Cholangiocarcinoma has one of the worst prognostic profiles of all cancers [1,2]. Bile duct biopsy and brush cytology are standard methods for obtaining pathological samples. However, these methods lack adequate sensitivity (8–67%) [3,4,5], and pathological evaluation, including comprehensive genome profiling (CGP), may be difficult owing to the small sample size. In contrast, bile can be easily collected by placing an endoscopic nasobiliary drainage (ENBD) tube in the same session as a bile duct biopsy. To overcome the diagnostic difficulties associated with cholangiocarcinoma, we previously attempted to collect large amounts of bile and malignant cells stored overnight for diagnosis.

We reported unexpected findings showing that cell block (CB) specimens from bile stored overnight contained more than 70 malignant glandular cells without degenerative changes per bile sample as a mean value [6]. While surgical intervention is administered to merely 35% of patients afflicted with this malignancy, chemotherapy regimens have been established as the standard of care [7,8]. However, their efficacy is moderate, and the number of regimens is limited. Currently, molecular-targeted drugs [9,10,11] and immune checkpoint inhibitors (ICI) are assuming a pivotal role in treatment, owing to the failure of standard chemotherapy [12,13]. CGP is a next-generation sequencing (NGS) approach that employs a single sample to assess hundreds of genes, as established in the guidelines [14]. As NGS requires sufficient tissue samples to obtain nucleic acids from tumor cells, it is not practically used for cholangiocarcinoma.

In this study, we aimed to use bile CB specimens for NGS of cholangiocarcinoma. First, we evaluated the feasibility of bile CB specimens for NGS of cholangiocarcinoma by comparing with the results of surgically resected specimens (SRS) prepared for the same patients. This is the first study to perform NGS of cholangiocarcinoma using CB specimens prepared from bile stored overnight.

## 2. Materials and Methods

### 2.1. Study Design, Patient Population, and Specimen Collection

This study was approved by the Institutional Review Board of the Gifu Municipal Hospital (Nos. 500 and 783) and registered at the University Hospital Medical Information Network Clinical Trial Registry (UMIN-CTR; UMIN000049618). Written informed consent was obtained from all patients before enrolment.

This study was conducted to evaluate the following in the CB and SRS, and compare the potential use of the CB specimen for identifying actionable and/or druggable gene(s) targets of cholangiocarcinoma: (1) the quantity and quality of nucleic acids, (2) the quality of NGS data, and (3) the number and the matched rate of genetic alterations.

A cohort of 64 Japanese patients who were prescribed endoscopic retrograde cholangiography (ERC) for biliary stricture diagnosis were prospectively recruited for this study from August 2018 to November 2022 at Gifu Municipal Hospital, Gifu, Japan. ERC was performed using a standard duodenoscope (JF-260V or TJF-260V; Olympus Medical Systems, Tokyo, Japan). A guidewire (0.025-inch M-ThroughTM; Asahi Intecc Co., Ltd., Aichi, Japan, or 0.025-inch VisiGlide2: Olympus Medical System Co., Ltd., Tokyo, Japan) was inserted into the biliary strictures. Subsequently, a 6-French ENBD tube (Gadelius Medical, Tokyo, Japan) was inserted into the obstructed bile duct to collect the bile (Figure 1A). The bile samples were stored in bottles overnight and collected the following morning for submission to the pathology department (Figure 1B). CB sections were prepared using the sodium alginate method as follows: (I) Bile obtained from the bottle was centrifuged at 1500 rpm for 10 min (Figure 1C); (II) Only the precipitate was collected and fixed in 10% neutral buffered formalin overnight (Figure 1D); (III) The resultant sample was centrifuged again at 1500 rpm for 10 min, and the precipitate was fixed in 10% neutral buffered formalin; (IV) Only the precipitate was collected, followed by the addition of 0.5 mL of 1% sodium alginate. The sample was agitated and centrifuged again at 1500 rpm for 10 min; (V) After discarding the supernatant, one or three drops of 1 M calcium chloride solution were added; and (VI) Finally, the sample was embedded in paraffin wax and processed to make 3–4 μm thick serial sections for HE staining (Figure 1E). Four pathologists (T.T., H.T., T.K., and N.W.) independently made histopathological and cytological diagnoses.

A total of 42 of the 64 enrolled patients had cholangiocarcinoma (65.6%, Appendix A), and the median of collected bile was 180 mL (range: 80–300 mL). Malignant cells were detected in the CB specimens of 28 patients. Eight patients underwent surgical resection for cholangiocarcinoma, and SRS was performed. Because the utility of bile CB specimens and their required number of cancer cells were unknown, and considering that the median cancer cell number of a bile CB specimen is 70 cells in 28 cases of cholangiocarcinoma (Appendix A), this study evaluated cases with large cancer cell counts (more than 100) in bile CB specimens (Figure 2). Patient characteristics, including age, sex, ERC findings, and the serum levels of CA19-9 and total bilirubin levels, are listed in Table 1.

Formalin-fixed paraffin-embedded (FFPE) blocks of the CB and SRS specimens were sliced into 20 serial sections (each with a thickness of 4 μm). Macrodissection was performed to achieve >20% estimated percentage tumor nuclei in resected bile duct specimens. Genomic DNA was extracted from ten consecutive FFPE sections sliced into specimens with a thickness of 5 µm using a QIAamp DNA FFPE Tissue Kit (Qiagen, Hilden, Germany) according to the manufacturer’s instructions. RNA was extracted from four consecutive FFPE sections sliced into specimens with a thickness of 10 µm using miRNeasy FFPE kit (Qiagen). Nucleic acid concentrations were measured on a Qubit 3.0 Fluorometer (Thermo Scientific, Paisley, UK) using Qubit dsDNA, RNA HS, and BR assay kits. Nucleic acid purity (quality) was evaluated based on the ratio of the absorbance at 260 nm/280 nm and 260 nm/230 nm. All authors had access to the study data and reviewed and approved the final manuscript.

### 2.2. Next-Generation Sequencing

The TruSight Oncology 500 (TSO500) was designed to analyze multiple biomarkers via both DNA and RNA sequencing derived from the CB and SRS pairs of the same patient. The number of short variants and genetic alterations, including mutations, CNV, splice variants and fusions, MSI, and TMB were evaluated.

The library preparation workflow of the Illumina TSO500 assay was used with DNA and RNA samples, according to the standard protocol. Briefly, adapter-ligated DNA and RNA were subjected to hybridization and target-capture steps, and libraries were amplified by polymerase chain reaction. Sequencing was performed using the Illumina NextSeq 2000 platform with 101 cycles of paired-end sequencing. Bioinformatics analyses, including evaluation of short variants, CNV, splice variants, fusion, MSI, and TMB, were performed using Illumina DRAGEN TSO500 Analysis Software v2.1.0. The NGS data quality was evaluated by assessing the mean number of reads as the mean base coverage.

### 2.3. Druggable or Cancer-Related Genetic Alterations

Targetable anticancer drugs for cholangiocarcinoma and cholangiocarcinoma-related genes are defined as druggable or cancer-related genetic alterations. Genetic alterations can be potentially targeted via using reported kinase inhibitors (KRAS, NRAS, ALK, EGFR, ERBB, FGFR1-3, ARID1A/B, CDKN2A/B, BAPI, RNF43, PIK3CA, BRAF, MET, PRKACA/B, NTRK, BRCA, RET, and ROS1), poly ADP-rinose polymerase inhibitors (ATM), and pharmacological inhibitors (IDH1/2) [15,16]. TMB and MSI have also been evaluated as markers of ICI. Cholangiocarcinoma-related genetic alterations (those reported in TP53, PTEN, ARID2, MLL, TERT, APOBEC, EPHA2, GNAS, ELF3, CTNNB1, AKT1, and SMAD4) have also been evaluated [17].

### 2.4. Statistical Analysis

All analyses were conducted using R v.4.0.2 (The R Foundation for Statistical Computing, Vienna, Austria). Fisher’s exact test was used for categorical variable analysis, and the Mann–Whitney U test was used for continuous variables. A *p*-value < 0.05 was considered statistically significant.

## 3. Results

### 3.1. Patient Characteristics and Endoscopic Findings

The patients included two women and one man aged 70 (case 1) and 76 (cases 2 and 3, Table 1). After endoscopic retrograde cholangiography (ERC), bile was obtained from all patients, and the volumes of bile used for the CB method were 100, 180, and 200 mL. The numbers of malignant cells on the glass slides of the CB specimens for cases 1–3 were 2785, 1024, and 200, respectively. All patients were diagnosed with distal cholangiocarcinoma using CB specimens and underwent pancreaticoduodenectomy.

### 3.2. Quantity and Quality of DNA and RNA Extracted Bile CB and Surgically Resected Specimens

DNA and RNA were extracted from the CB and SRS pairs of three patients. The concentration of DNA in each pair of CB and SRS were as follows: [case 1: 66.6 and 182.7 ng/mL], [case 2: 10.3 and 110.6], and [case 3: 4.1 and 163.0] (*p* < 0.01). The amounts of the DNA extracted from CB and SRS were as follows [2665.3 and 7309.0 ng], [413.3 and 4424.0], and [162.7 and 6521.3] (*p* < 0.01), respectively, for cases 1–3. The concentrations of RNA were [49.1 and 415.1 ng/mL], [18.3 and 297.8], and [6.7 and 231.2] (*p* < 0.01), respectively, for cases 1–3. The extracted RNA amounts were [982.0 and 8301.3 ng], [366.7 and 5956.7], and [160.8 and 5548.0], respectively, for cases 1–3 (*p* < 0.01). The concentrations and amounts of DNA and RNA were significantly lower in CB specimens than in SRSs. No significant differences were noted in the quality of the DNA and RNA, including the ratios of the absorbance at 260 nm/230 nm and 260 nm/280 nm (Table 2).

### 3.3. Total Number of Reads in NGS and NGS Quality

The total number of reads in the DNA of each pair of CB and SRS were as follows: [case 1: 94,673,584 and 90,863,546], [case 2: 95,077,806 and 102,191,580], and [case 3: 81,323,720 and 93,161,788] (*p* = 1.0), respectively. The mean x0.4 coverages were [96.6 and 92.7], [96.5 and 94.9], and [96.3 and 95.3], respectively, for cases 1–3 (*p* = 0.98). The total number of RNA reads from each pair were [22,101,328 and 9,345,634], [24,342,318 and 24,806,180], and [19,600,362 and 25,684,972] (*p* < 0.01), respectively, for cases 1–3. The mean coverage was [2508.0 and 411.2], [4616.1 and 4743.1], and [2780.0 and 4963.1] (*p* < 0.01), respectively, for cases 1–3. Although there were no significant differences in the total number of reads or mean coverage of DNA, the total number of reads and mean coverage of RNA were significantly higher in the CB specimens than in the SRSs.

### 3.4. Short Variant, CNV, Splice Variants, and Fusion

TSO500 was used to evaluate the short variants in all the specimens. The SNV loads for each pair of CB and SRS were as follows: [case 1: 1111 and 1126], [case 2: 1485 and 1097], and [case 3: 1041 and 1041] (*p* = 0.66). The SNV loads were similar in cases 1 and 3. However, in case 2, the CB SNV loads were higher than those of SRS. The number and rate of base substitution subtypes (e.g., A>C, A>G…) in each pair are shown in Table 2. No significant difference was observed in the rate of base substitutions between CB and SRS for each pair. The total numbers of short InDels and MNV were [72 and 93], [91 and 64], and [88 and 66] (*p* = 0.52), respectively, for cases 1–3. No significant differences were observed in the rates of short InDels and MNV for each pair (Figure 3A,B). The total number of short variants in each pair were [1183 and 1219], [1576 and 1161], and [1129 and 1107] (*p* = 0.51), respectively, for cases 1–3.

The short variant matching rate for each pair was 96.4% (case 1, the number of matched CB and SRS short variant/number of all short variants: 1165/1208), 72.7% (case 2, 1152/1585), and 95.2% (case 3, 1091/1145). The coverage rates of short variants of the SRS based on short variants of CB specimen were 97.9% (case 1, number of the covered CB/number of all SRS short variants: 1165/1190), 99.2% (case 2, 1152/1161), and 98.6% (case 3, 1091/1107). The short variants in cases 1 and 3 matched at a high rate. In case 2, the short variants of the CB specimen were mostly covered by the short variant of SRS and could be higher than the SRS short variant number (Table 3, Figure 4).

None of the specimens exhibited CNVs. The numbers of splice variants in each pair of CB and SRS are listed in Table 2.

### 3.5. Genetic Alterations

TSO500 can evaluate 588 cancer-relevant genetic alterations. Because CNVs were not detected in any of the specimens, only short variants were associated with genetic alterations. Figure 5A summarizes the relationship between the short variants and genetic alterations. Mutations caused by short variants were observed in each CB and SRS pair. The number of missense mutations were as follows: [case 1: 83 and 84], [case 2: 90 and 79], and [case 3: 80 and 78]. The numbers of synonymous codings were [151 and 151], [170 and 145], and [150 and 150], respectively, for cases 1–3. The numbers of nonsense mutations were [0 and 0], [2 and 1], and [0 and 1], and those of frameshift mutations were [1 and 0], [0 and 0], and [1 and 0], respectively, for cases 1–3. No significant difference was observed in mutation rates between the CB and SRS groups (Table 2; Figure 3C).

The numbers of genetic alterations in each pair were [214 and 214], [245 and 217], and [212 and 211], respectively. No significant difference was observed in the number of genetic alterations between the pairs of specimens. The gene matching rates of CB and SRS in each pair were 97.6% (case 1, the number of matched CB and SRS genes [genetic alterations and wild-type]/the number of all evaluated genes: 584/588), 89.5% (case 2, 556/588), and 98.1% (case 3, 583/588). The coverage rates of all SRS genes based on CB specimen genes were 99.7% (case 1, 584/586), 99.6% (case 2, 556/558), and 99.7% (case 3, 583/585). Most of the evaluated CB specimens comprised SRS genes (Table 3, Figure 4).

### 3.6. Druggable or Cancer-Related Genetic Alterations, MSI, and TMB

Forty-three genes were evaluated for druggable or cancer-related genetic alterations in all specimens. Figure 5B,C summarize the relationships between short variants and druggable or cancer-related genetic alterations. The numbers of genetic alterations of CB and SRS were as follows: [case 1: 27 and 28], [case 2: 27 and 26], and [case 3: 22 and 22], respectively. No significant differences were observed between groups CB and SRS (Table 3).

MSI and TMB were successfully evaluated using NGS for all six specimens (Table 2). The percentages of unstable MSI sites for each pair of CB and SRS were as follows: [case 1: 1.67 and 0.90], [case 2: 1.61 and 2.40], and [case 3: 12.5 and 2.44], respectively. MSI results showed that all specimens were stable. The total TMB values for each pair were [2.35 and 9.48], [8.60 and 2.35], and [0 and 1.56], respectively, for cases 1–3. The results showed no high TMB values.

## 4. Discussion

The ERC is required for the treatment and diagnosis of patients with biliary strictures. Bile is easily collected during ERC or ENBD for biliary drainage. Recently, Arechederra et al. developed a mutational analysis method for bile collected during ERCP, called the Bilemut assay, using an NGS panel open for clinical laboratory implementation [18]. Their analysis of 30 paired tissue samples and cell-free DNA (cfDNA) of bile showed increased diagnostic sensitivity and number of mutations detected compared to the corresponding paired tissue samples. While they used fresh bile samples for NGS analysis, our analysis was conducted using CB prepared from bile stored overnight. Initially, we speculated that bile stored overnight may mainly contain degenerative tumor cells and that cfDNA is not sufficient for NGS analysis. However, our NGS results showed the presence of cancer-related genetic alterations in bile CB specimens, similar to the SRS samples (Figure 4 and Figure 5). Taken together, employing bile for NGS in cholangiocarcinoma cases is a logical approach, particularly when acquiring tissue specimens is typically challenging. However, it is important to consider whether cfDNA is related to cancer cells or other cells such as inflammatory cells.

Although the amount of nucleic acid extracted from the CB specimens was lower than that from SRS, the number of reads and coverage of NGS was comparable to that of SRS. In general, RNA extraction is more difficult than DNA extraction because of RNA instability. However, the quality and number of RNA reads in CB were similar to or greater than those in SRS. The high quality of nucleic acids in CB may be related to the difference in the required penetration time of formalin compared with that of SRS. Although degeneration of malignant cells is inevitable due to bile exposure, centrifugation clearly separates malignant cells from degenerated cells due to the difference in specific gravity. We were able to prepare CB specimens that accumulated malignant cells from the tiny precipitate, in which 10% buffered formalin quickly penetrates. The coverage rates (97.9–99.2%) of short variant and genes in the SRS based on the CB specimens were quite high, suggesting that the nucleic acids extracted from CB were of high quality and can be evaluated as with SRS. Among the cases we tested, the short variant and gene matching rates (95.2–98.1%) were quite high in cases 1 and 3. However, the matching rates were low in case 2 because the number of short variants of the CB specimen was higher than that of the SRS specimen (Figure 4A). In case 2, the CB specimen covered almost all variants of paired SRS, suggesting that the nucleic acids in SRS might have been damaged, potentially resulting in the absence of pertinent genetic data, possibly due to excessive fixation time (over 72 h). Based on these findings, we concluded that CB specimens were superior to SRSs for evaluating genetic alterations.

The evaluation of RNA, MSI, and TMB is important for identifying potential anticancer therapies [9,10,11,12,13,19]. In this study, cancer tissues resected immediately after surgery in our cases were suggested to be microsatellite-stable (MSS) and TMB-low subtypes. The findings of NGS analyses of the CB and SRS confirmed these results and suggested that the CB specimens contained sufficient DNA and RNA from cancer cells to evaluate MSI and TMB.

This study had some limitations. As both CB specimens and SRS were limited, the number of cases in this study was also limited. Our study evaluated only cases of extrahepatic cholangiocarcinoma. Nonetheless, it is important to evaluate hilar, intrahepatic cholangiocarcinoma, and gallbladder cancer cases; however, five of the eight surgical cases were extrahepatic cholangiocarcinoma, and the remaining cases were one case of intrahepatic cholangiocarcinoma and two cases of gallbladder cancer, which was a very small number of cases. In addition, at the beginning of this study, it was unknown whether bile CB would be useful. Therefore, as a subsequent challenge to NGS analysis, we evaluated limited cases with more than 100 malignant cells in CB specimens prepared from bile stored overnight, which included three extrahepatic cholangiocarcinoma cases. However, as this was a small-number study, various types of cholangiocarcinoma (intrahepatic, hilar, extrahepatic cholangiocarcinoma, and gallbladder cancer) are needed to confirm the accuracy of bile CB specimens. There is also a need to evaluate the minimum number of cancer cells and the amount of collected bile required for NGS testing, and whether NGS tests can be performed on bile CB specimens without cancer cells using cfDNA. Second, although TSO 500 fully covered the genes tested by CGP and also provided information about MSI and TMB, we did not perform whole-genome sequencing. In the future, it will be imperative to subject such specimens to comprehensive whole-genome sequencing as a fundamental research step. This is crucial to provide tailored therapeutic choices for specific patients dealing with cholangiocarcinoma.

The aim of this study was to evaluate the feasibility of bile CB for NGS-based CGP. The findings demonstrate that bile CB is sufficient for CGP (Figure 5B), and the cancer-associated genetic alterations observed in CB specimens were consistent with those of SRSs (Figure 5C). Additionally, the CB specimens have the potential to serve as valuable resources for fundamental or oncologic research investigations, as we can obtain a large amount of bile at any time during ENBD. Furthermore, CB specimens can be stored for a long period as FFPE samples and tested whenever CGP becomes necessary.

## 5. Conclusions

The NGS results of bile CB specimens fully covered the variants and genetic alterations observed in paired SRS samples. CB from bile stored overnight is easy to prepare and can be used as a novel specimen type for the NGS-based evaluation of cholangiocarcinoma. However, as this was a small-number study, additional cases are needed to confirm the accuracy of bile CB specimens.

## Figures and Tables

**Figure 1 cells-13-00925-f001:**
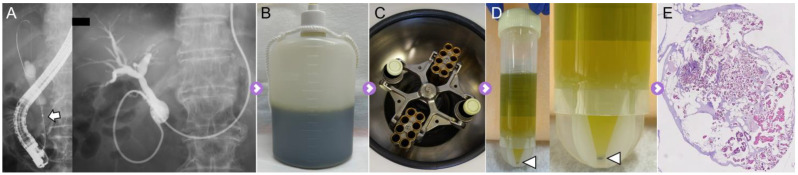
**Cell block specimens prepared using overnight-stored bile.** (**A**) The malignant biliary stricture is detected in the distal common bile duct (arrow). ENBD tube is placed in the occluded bile duct via the biliary stricture. (**B**) All bile from the ENBD tube was stored overnight. (**C**) All of the stored bile was used for the development of the CB specimen. Bile was processed by centrifugation for 10 min. (**D**) Only the precipitate was collected, followed by filling with 10% neutral buffered formalin (arrowhead). Finally, the specimens were embedded in paraffin wax for creating FFPE sections. (**E**) The FFPE blocks of CB specimen were evaluated by HE staining and used for the extraction of DNA and RNA.

**Figure 2 cells-13-00925-f002:**
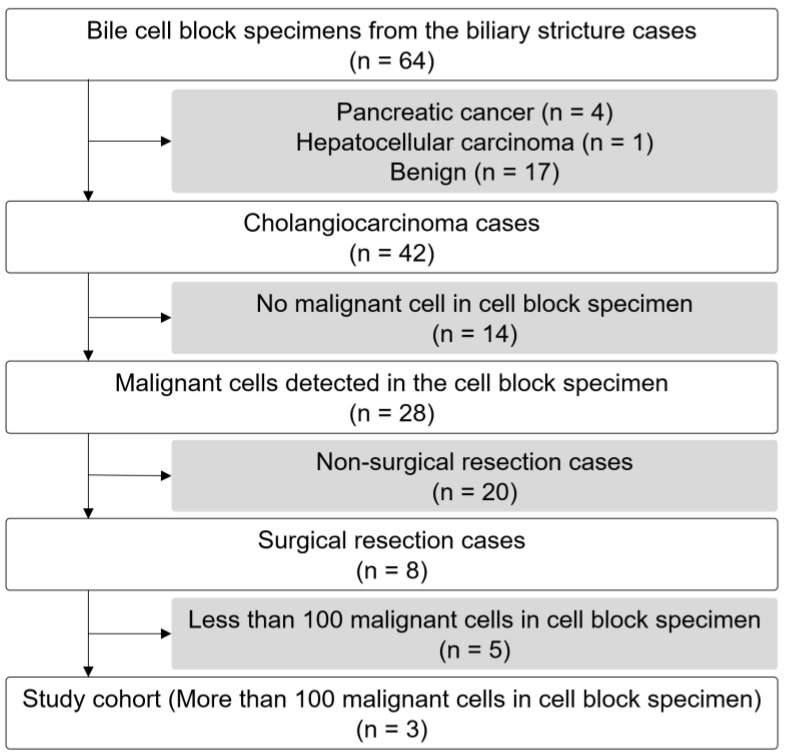
Flowchart of patient selection.

**Figure 3 cells-13-00925-f003:**
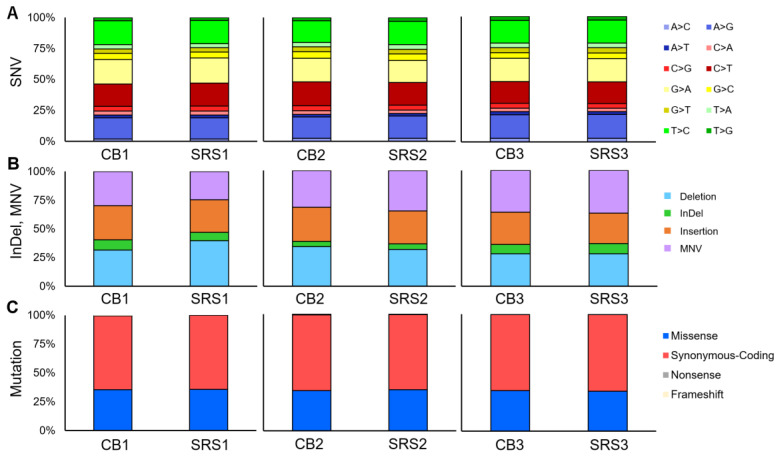
The proportion of short variants and mutations detected by TruSight Oncology 500 DNA panel assay. (**A**) The proportion of SNV type. (**B**) The proportion of InDel and MNV. (**C**) The proportion of the type of mutation.

**Figure 4 cells-13-00925-f004:**
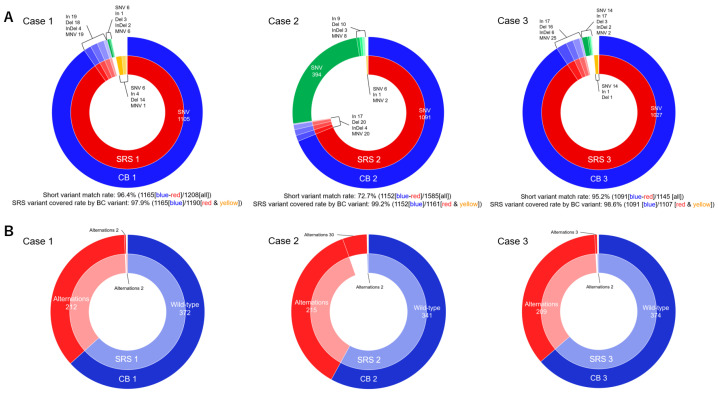
Compatibility of short variants and genetic alterations of CB and SRS. (**A**) CB and SRS in case 1: Short variant matched rate: 96.4% (1165 [blue-red]/1208 [all]). The coverage rate of SRS variants by CB variants: 97.9% (1165 [blue]/1190 [red-yellow]). Case 2: Short variant matching rate: 72.7% (1152 [blue-red]/1585 [all]). The coverage rate of SRS variants by CB variants: 99.2% (1152 [blue]/1161 [red-yellow]). Case 3: Short variant matching rate: 95.2% (1091 [blue-red]/1145 [all]). The coverage rate of SRS variants by CB variants: 98.6% (1091 [blue]/1107 [red-yellow]). (**B**) CB and SRS in Case 1: The gene matched rate: 97.6% (584/588). The coverage rate of all SRS genes based on CB specimen genes: 99.7% (584/586). Case 2: The gene matching rate: 89.5% (556/588). The coverage rate of all SRS genes based on CB specimen genes: 99.6% (556/558). Case 3: The gene matching rate: 98.1% (583/588). The coverage rate of all SRS genes based on CB specimen genes: 99.7% (583/585).

**Figure 5 cells-13-00925-f005:**
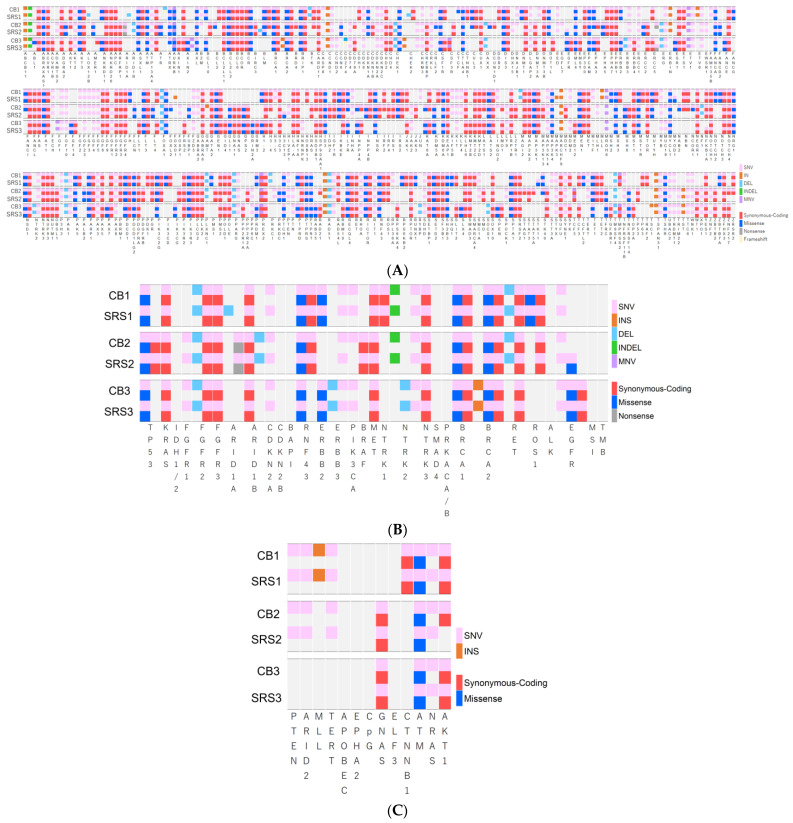
(**A**) Association between short variants and genetic alterations. (**B**) Association between short variants and the druggable genetic alterations. (**C**) Association between short variants and cancer-related genetic alterations.

**Table 1 cells-13-00925-t001:** Basic characteristics of the patients.

Case No.	Case 1	Case 2	Case 3
Gender	F	F	M
Age, years	70	76	76
T-Bil, mg/dL	17.2	12.8	0.8
CA 19-9, U/mL	95.7	620.3	10
Location of stricture	Distal	Distal	Distal
Malignant biliary obstruction length, mm	43	23	14
Volume of collected bile, mL	100	180	200
Malignant cell number of Cell block specimen, n	2785	1024	200
Pathological diagnosis	Distal cholangiocarcinoma	Distal cholangiocarcinoma	Distal cholangiocarcinoma
UICC Stage	IB	IB	IIA

**Table 2 cells-13-00925-t002:** The details of extracted DNA and RNA and NGS (TruSight Oncology 500 assay) of each specimen.

	Case 1			Case 2			Case 3		
	CB1	SRS1		CB2	SRS2		CB3	SRS3	
**Extracted DNA**									
DNA consentration, ng/µL	66.6	182.7		10.3	110.6		4.1	163.0	
DNA amount, ng	2665.3	7309.0		413.3	4424.0		162.7	6521.3	
A260/A280	1.883	1.903		1.887	2.087		1.730	1.997	
A260/A230	1.990	0.843		1.153	0.595		0.280	2.080	
Total number of reads in NGS, n	94,673,584	90,863,546		95,077,806	102,191,580		81,323,720	93,161,788	
Mean x0.4 coverage	96.6	92.7		96.5	94.9		96.3	95.3	
**Extracted RNA**									
RNA consentration, ng/µL	49.1	415.1		18.3	297.8		6.7	231.2	
RNA amount, ng	982.0	8301.3		366.7	5956.7		160.8	5548.0	
A260/A280	1.807	1.837		1.810	2.043		1.620	2.023	
A260/A230	1.380	1.580		1.666	2.113		0.953	2.167	
Total reads of reads NGS, n	22,101,328	9,345,634		24,342,318	24,806,180		19,600,362	25,684,972	
Mean coverage	2508.0	411.2		4616.1	4743.1		2780.0	4963.1	
**DNA analysis**			*p*-value			*p*-value			*p*-value
**SNV, n (%)**	1111 (100%)	1126 (100%)		1485 (100%)	1097 (100%)		1041 (100%)	1041 (100%)	
A>C	21 (1.9%)	21 (1.9%)	1.0	36 (2.4%)	26 (2.4%)	1.0	28 (2.7%)	29 (2.8%)	1.0
A>G	189 (17.0%)	191 (17.0%)	1.0	255 (17.2%)	197 (18.0%)	0.60	198 (19.0%)	199 (19.1%)	0.91
A>T	25 (2.2%)	26 (2.3%)	1.0	32 (2.2%)	22 (2.0%)	0.89	24 (2.3%)	23 (2.2%)	1.0
C>A	35 (3.2%)	36 (3.2%)	1.0	42 (2.8%)	30 (2.7%)	0.90	30 (2.9%)	28 (2.7%)	0.89
C>G	44 (4.0%)	46 (4.0%)	0.91	62 (4.2%)	45 (4.1%)	1.0	39 (3.7%)	39 (3.7%)	1.0
C>T	201 (18.1%)	208 (18.5%)	0.83	286 (19.3%)	201 (18.3%)	0.58	182 (17.5%)	180 (17.3%)	0.95
G>A	219 (19.7%)	229 (20.3%)	0.75	284 (19.1%)	197(18.0%)	0.47	193 (18.5%)	193 (18.5%)	1.0
G>C	55 (5.0%)	54 (4.8%)	0.92	78 (5.2%)	58 (5.3%)	0.93	46 (4.4%)	46 (4.4%)	1.0
G>T	41 (3.7%)	40 (3.6%)	0.91	61 (4.1%)	41 (3.7%)	0.68	43 (4.1%)	45 (4.3%)	0.91
T>A	38 (3.4%)	36 (3.2%)	0.81	50 (3.4%)	41 (3.7%)	0.67	37 (3.6%)	38 (3.7%)	1.0
T>C	215 (19.3%)	212 (18.8%)	0.79	262 (17.6%)	207 (18.9%)	0.44	189 (18.2%)	191 (18.4%)	0.95
T>G	28 (2.5%)	27 (2.4%)	0.89	37 (2.5%)	32 (2.9%)	0.54	32 (3.1%)	30 (2.9%)	0.90
**InDel, MNV, n (%)**	72 (100%)	93 (100%)	*p*-value	91 (100%)	64 (100%)	*p*-value	88 (100%)	66 (100%)	*p*-value
Deletion	21 (29.2%)	32 (34.4%)	0.51	30 (33.0%)	20 (31.2%)	0.86	19 (21.6%)	17 (25.7%)	0.57
Insertion	20 (27.8%)	23 (24.7%)	0.72	26 (28.5%)	18 (28.1%)	1.0	34 (38.6%)	18 (27.3%)	0.17
InDel	6 (8.3%)	18 (19.4%)	0.07	7 (7.7%)	4 (6.3%)	1.0	8 (9.1%)	6 (9.1%)	1.0
MNV	25 (34.7%)	20 (21.5%)	0.08	28 (30.8%)	22 (34.4%)	0.73	27 (30.7%)	25 (37.9%)	0.39
**Mutation, n (%)**	235 (100%)	235 (100%)	*p*-value	262 (100%)	225 (100%)	*p*-value	231 (100%)	229 (100%)	*p*-value
Missense	83 (35.3%)	84 (35.7%)	1.0	90 (34.3%)	79 (35.1%)	0.92	80 (34.6%)	78 (34.1%)	0.92
Synonymous-Coding	151 (64.3%)	151 (64.3%)	1.0	170 (64.9%)	145 (64.4%)	0.92	150 (64.9%)	150 (65.5%)	0.92
Nonsense	0 (0%)	0 (0%)	N.A.	2 (0.8%)	1 (0.5%)	1.0	0 (0%)	1 (0.4%)	0.50
Frameshift	1 (0.4%)	0 (0%)	1.0	0 (0%)	0 (0%)	N.A.	1 (0.5%)	0 (0%)	0.50
CNV, n	0	0		0	0		0	0	
Percent unstable MSI sites	1.67	0.90		1.61	2.40		12.5	2.44	
Total TMB	2.35	9.48		8.60	2.35		0	1.56	
**RNA analysis**									
Number of splice variant, n	0	0		1	1		0	1	
Number of fusion, n	3	0		3	0		1	0	

**Table 3 cells-13-00925-t003:** Compatibility of short variants and genetic alterations in CB and SRS.

	Case 1			Case 2			Case 3		
	CB1	SRS1		CB2	SRS2		CB3	SRS3	
**Short variant**									
Total number of short variant, n	1183	1219		1576	1161		1129	1107	
Number of unmatched short variant, n	18	25		424	9		38	16	
Short variant matched rate, %	96.4%			72.7%			95.2%		
SRS variant covered rate by CB variant, %	97.9%			99.2%			98.6%		
**Total evaluated genetic alternation in TSO 500 (n = 588)**			*p*-value			*p*-value			*p*-value
Number of genetic alteration, n (%)	214 (36.4%)	214 (36.4%)	1.0	245 (41.7%)	217 (36.9%)	0.11	212 (36.1%)	211 (35.9%)	1.0
Number of unmatched genetic alteration, n	2	2		2	30		2	3	
Gene matched rate, %	97.6%			89.5%			98.1%		
SRS gene covered rate by CB gene, %	99.7%			99.6%			99.7%		
**Druggable or cancer-related genetic alternation (n = 43)**			*p*-value			*p*-value			*p*-value
Number of genetic alteration, n (%)	27 (62.8%)	28 (65.1%)	1.0	27 (62.8%)	26 (60.5%)	0.83	22 (51.2%)	22 (51.2%)	1.0
Number of unmatched genetic alteration, n	1	1		1	1		0	0	
Gene matched rate, %	97.7%			97.7%			100%		

## Data Availability

The dataset is available on request from the authors.

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
