# Peer review of "Possibility of Cell Block Specimens from Overnight-Stored Bile for Next-Generation Sequencing of Cholangiocarcinoma"

_cells, 2024, doi:10.3390/cells13110925_

Round 1
Reviewer 1 Report
Comments and Suggestions for Authors
This study demonstrated that NGS analysis using Bile's cell block can effectively represent the molecular characteristics of cancer tissue. Although the research process was systematically and thoroughly described and contains some novel elements, the following areas require improvement.
1. In the bile cell block of 42 cholangiocarcinoma patients, malignant cells were detected in only 28 patients, and analysis was possible in only 66.7% of them. Among the eight bile cell block samples that could be directly compared to the surgical specimen, only three blocks (37.5%) contained more than 100 malignant cells that were actually analyzed. These data indicate that the percentage of bile cell blocks usable for NGS in cholangiocarcinoma is only up to 66.7%, which seems to be an insufficient rate. Although the feasibility of using bile cell blocks is acknowledged, their accuracy is questionable.Additionally, the limited analysis of three cases is insufficient to draw conclusions.
2. Why were samples with fewer than 100 malignant cells excluded from the cell block specimen? What criteria were used to set this cutoff value?
3. PTBD or ERCP must be performed to collect bile. During this process, a tissue biopsy can be performed, and NGS can be conducted on the obtained tissue. What are the advantages of NGS testing using a bile cell block over NGS using tissue obtained by procedure?
4. The sample in which many malignant cells are detected in the cell block is likely to be at an advanced stage. What is the stage information?
Author Response
Reviewer 1
This study demonstrated that NGS analysis using Bile's cell block can effectively represent the molecular characteristics of cancer tissue. Although the research process was systematically and thoroughly described and contains some novel elements, the following areas require improvement.
- In the bile cell block of 42 cholangiocarcinoma patients, malignant cells were detected in only 28 patients, and analysis was possible in only 66.7% of them. Among the eight bile cell block samples that could be directly compared to the surgical specimen, only three blocks (37.5%) contained more than 100 malignant cells that were actually analyzed. These data indicate that the percentage of bile cell blocks usable for NGS in cholangiocarcinoma is only up to 66.7%, which seems to be an insufficient rate. Although the feasibility of using bile cell blocks is acknowledged, their accuracy is questionable. Additionally, the limited analysis of three cases is insufficient to draw conclusions.
Response.
Thank you for your kind review.
The reason and purpose of this study is that the success rate of biopsy specimen collection is insufficient (8–67%), while bile can be collected in sufficient quantities in cholangiocarcinoma; therefore, we investigated whether bile could replace biopsy specimens for NGS testing. In addition, the CB specimen can be created as FFPE, which can be stored for a long period (2–3 years). We evaluated its utility as a bile CB specimen.
However, there have been no previous studies on whether NGS can be performed on bile CB samples or the number of cancer cells required. Therefore, we compared three cases with a large number of cancer cells, which could be compared with surgical specimens. We had favorable results in the CB specimens of these three cases; however, as the reviewer pointed out, this was a study with a small number of cases. In the future, it will be necessary to evaluate the minimum number of cancer cells required for NGS testing and whether NGS tests can be performed on bile CB samples without cancer cells using cfDNA. We have added this point to lines 306-319 and revised our conclusion (lines 336-337).
- Why were samples with fewer than 100 malignant cells excluded from the cell block specimen? What criteria were used to set this cutoff value?
Response.
As mentioned in Response 1, the utility of bile CB specimens and the required number of cancer cells in bile CB specimens were unknown; therefore, this study was conducted using cases with large cancer cell counts in bile CB specimens. Because the median number of cancer cells in bile CB specimens was 70 in a previous study (Okuno et al. Cancers 2022,14, 2701) and 28 patients with cholangiocarcinoma from whom cancer cells were obtained (Supplemental Table 1), we conducted this research based on more than 100 cases. However, the minimum number of cancer cells required for NGS needs to be determined in additional cases. We have added this point to lines 99-103 and 311-314.
- PTBD or ERCP must be performed to collect bile. During this process, a tissue biopsy can be performed, and NGS can be conducted on the obtained tissue. What are the advantages of NGS testing using a bile cell block over NGS using tissue obtained by procedure?
Response.
Thank you for pointing this out. The sensitivity of transpapillary biopsy is insufficient (8-67%), and even if cancer tissue can be obtained, NGS cannot be performed using biopsy specimens because of their small size.
In contrast, bile can be easily collected by placing ENBD in the same session as the biliary biopsy. If NGS (CGP) can be performed on bile CB specimens, there will be more opportunities to perform cholangiocarcinoma CGP tests and improve cholangiocarcinoma prognosis.
In addition, using bile as the CB specimen instead of raw (fresh) bile, it can be stored for as long as FFPE and tested whenever NGS tests become necessary. Therefore, we believe that the bile CB method can enable NGS testing for cholangiocarcinoma.
We have added this point to lines 34-38 and 330-331.
- The sample in which many malignant cells are detected in the cell block is likely to be at an advanced stage. What is the stage information?
Response.
We have added information on the UICC stages for the three cases in Table 1. We also added information on the 42 cholangiocarcinoma cases, including the UICC stage, as shown in Supplemental Table 1.
Reviewer 2 Report
Comments and Suggestions for Authors
The author Okuno et al reported a potential use of bile cell block for NGS based evaluation of cholangiocarcinoma. The authors showed the NGS data of bile cell block is similar as surgically resected specimen NGS data. all 3 cases are all distal cholangiocarcinoma, and the malignant biliary obstruction length is from 14 mm to 43 mm, the collected bile volume is 100-200 ml. the NGS results are promising , however, there are some comments:
(1). Have you ever checked other type of cholangiocarcinoma type: hilar cholangiocarcinoma, Intrahepatic cholangiocarcinoma? Intrahepatic cholangiocarcinoma are more common cases.
(2). what is the lowest volume of bile can be used for cell block NGS? For most cases of ERCP procedure, it is hard to collect that big volume of 100 -200 ml bile for NGS.
(3). the authors only did bile cell block NGS for 3 cases, more cases are required to get convinced conclusion.
Author Response
Reviewer 2
The author Okuno et al reported a potential use of bile cell block for NGS based evaluation of cholangiocarcinoma. The authors showed the NGS data of bile cell block is similar as surgically resected specimen NGS data. All 3 cases are all distal cholangiocarcinoma, and the malignant biliary obstruction length is from 14 mm to 43 mm, the collected bile volume is 100-200 ml. the NGS results are promising, however, there are some comments:
(1). Have you ever checked other type of cholangiocarcinoma type: hilar cholangiocarcinoma, Intrahepatic cholangiocarcinoma? Intrahepatic cholangiocarcinoma are more common cases.
Response.
Thank you for your comments. It was necessary to determine whether bile CB specimens had genetic results comparable to those of the surgical specimens. Therefore, it is essential to evaluate cases from which surgical specimens are obtained.
It is important to evaluate hilar, intrahepatic cholangiocarcinoma, and gallbladder cancer; however, five of the eight surgical cases involved extrahepatic cholangiocarcinoma. The remaining cases were one case of intrahepatic cholangiocarcinoma and two cases of gallbladder cancer, which was a small number of cases.
In addition, when we initiated this study, the required number of cancer cells in bile CB specimens was unknown. Therefore, we used the CB specimen with a large number of cancer cells (more than 100). Therefore, all cases in this study were extrahepatic cholangiocarcinomas.
We have added this limitation to 305-314. We have also added that because favorable results were obtained in these three cases; thus, it is necessary to evaluate hilar, intrahepatic cholangiocarcinoma, and gallbladder cancer in the future, as mentioned in lines 314-319.
(2). what is the lowest volume of bile can be used for cell block NGS? For most cases of ERCP procedure, it is hard to collect that big volume of 100 -200 ml bile for NGS.
Response.
Thank you for your question regarding the creation of the BCB method. In our study cohort, the lowest bile volume in cholangiocarcinoma cases was 80 mL. Although CB specimens can be created from a small bile volume, the volumes of nucleic acids (DNA and RNA) can be reduced owing to the small bile volume. Therefore, our study was performed using bile stored overnight to collect additional bile. The median bile volume in this study was 180 (80-300) ml. We have added this information to Supplemental Table 1 and lines 39-40 and 96-97.
(3). the authors only did bile cell block NGS for 3 cases, more cases are required to get convinced conclusion.
Response.
Thank you for the valuable review. Because there was no previous study on whether NGS could be performed on bile CB samples and the required number of cancer cells in bile CB specimens for NGS tests, we compared the cases in which we were able to contrast the surgical-resected and CB specimen with a large number of cancer cells. Therefore, this study evaluated three cases.
However, we agree that more cases should be evaluated to reach convincing conclusions. In the future, it will also be necessary to evaluate the minimum number of cancer cells required for NGS testing and whether NGS tests can be performed on bile CB samples without cancer cells using cfDNA. We have added this point to lines 305-319 and revised our conclusion (lines 336-337).
Reviewer 3 Report
Comments and Suggestions for Authors
With this manuscript, Authors aimed to demonstrate that CB specimens, i.e. cell blocks obtained from bile stored overnight are suitable for performing NGS analyses on both DNA and RNA for the identification of cancer related genes of cholangiocarcinoma, which is essential for the identification of anticancer therapies. They show an enormous amount of NGS data to demonstrate that CB specimens are equally suitable for genetic analyses as surgically resected specimens (SRS). The Authors conclude that because bile CB is easy to prepare in general hospitals, their results suggest the potential use of bile CB as a novel method for NGS-based evaluation of cholangiocarcinoma. The methods used and results presented are convincing, however the positive results of the study face some major limitations that, in my view, limit its innovative potential. Of 42 CB specimens collected from patients with confirmed cholangiocarcinoma, slightly more than half presented malignant cells. Of the eight CB specimens for which SRS specimens were available, only three had sufficient numbers of malignant cells (>100) to conduct NGS. By the way, Authors do not tell us how many of the total CB samples had malignant cell numbers > 100. Therefore, to demonstrate the suitability of CB samples for diagnostic use in clinical settings, only three paired samples were examined. This is too small a number of samples to ensure the feasibility and accuracy of biliary CB for NGS-based CG.
Author Response
Reviewer 3
With this manuscript, Authors aimed to demonstrate that CB specimens, i.e. cell blocks obtained from bile stored overnight are suitable for performing NGS analyses on both DNA and RNA for the identification of cancer related genes of cholangiocarcinoma, which is essential for the identification of anticancer therapies. They show an enormous amount of NGS data to demonstrate that CB specimens are equally suitable for genetic analyses as surgically resected specimens (SRS). The Authors conclude that because bile CB is easy to prepare in general hospitals, their results suggest the potential use of bile CB as a novel method for NGS-based evaluation of cholangiocarcinoma. The methods used and results presented are convincing, however the positive results of the study face some major limitations that, in my view, limit its innovative potential. Of 42 CB specimens collected from patients with confirmed cholangiocarcinoma, slightly more than half presented malignant cells. Of the eight CB specimens for which SRS specimens were available, only three had sufficient numbers of malignant cells (>100) to conduct NGS. By the way, Authors do not tell us how many of the total CB samples had malignant cell numbers > 100. Therefore, to demonstrate the suitability of CB samples for diagnostic use in clinical settings, only three paired samples were examined. This is too small a number of samples to ensure the feasibility and accuracy of biliary CB for NGS-based CGP.
Response.
Thank you for your kind review.
Although the sensitivity of transpapillary biopsy is insufficient, bile can easily be collected via ENBD placement. If NGS tests can be performed on bile CB specimens, there will be more opportunities to perform cholangiocarcinoma CGP testing and improve prognosis. However, no studies have investigated whether NGS can be performed on bile CB samples. In addition, the minimum number of cancer cells required for NGS remains unclear. Therefore, we compared three cases with a large number of cancer cells, which could be compared with surgical specimens.
In addition, because the median number of cancer cells in bile CB specimens was 70 in 28 patients with cholangiocarcinoma from whom cancer cells were obtained (Supplemental Table 1), we conducted this research based on more than 100 cases.
Although the three cases studied showed favorable results, this was a small case study. In the future, it will be necessary to evaluate the minimum number of cancer cells required for NGS testing and whether NGS tests can be performed on bile CB samples without cancer cells using cfDNA. We have added this point to lines 305-319 and revised our conclusion (lines 336-337).
Round 2
Reviewer 1 Report
Comments and Suggestions for Authors
Thank you for your response to my review. I consider the answers to questions 2 and 4 to be sufficient. Unfortunately, the responses to questions 1 and 3 are not deemed sufficient, but I think that the answers provided by the authors are the best possible within the scope of this study.
Author Response
Thank you for your kind review. We are relieved to receive the comment that we have adequate responses for 2 and 4 comments. However, we apologize to not have enough responses to 1 and 3 comments. In particular, because of the small number study, we agree that additional cases should be considered in the future.
Our results suggest the usefulness of bile CB specimens, which can be prepared simply by collecting bile; however, we need to increase the number of cases and evaluate the condition of bile CB specimens for CGP, including the number of cancer cells and the amount of collected bile, in the future. We added this point in lines 317-318.
Reviewer 3 Report
Comments and Suggestions for Authors
Although the authors added some statements in the background section and conclusion that somehow further acknowledged the limitations of their work, my concerns remain. But, wanting to emphasize the originality of this experimental work, which I recognize is well conducted for the NGS analysis part, I suggest removing the word "accuracy" from lines 53 and 326 of the text, and adding the word "possible" to the title.
Author Response
Thank you for your kind review and comments. We appreciate for recognizing our study as an original experimental study, although there are some limitations, including the small number of cases.
We remove “accuracy” from lines 53 and 326. The title is also revised as “Possibility of cell block specimens from overnight-stored bile for next-generation sequencing of cholangiocarcinoma”.